# The Role of Insulin-like Peptide in Maintaining Hemolymph Glucose Homeostasis in the Pacific White Shrimp *Litopenaeus vannamei*

**DOI:** 10.3390/ijms23063268

**Published:** 2022-03-17

**Authors:** Manwen Su, Xiaojun Zhang, Jianbo Yuan, Xiaoxi Zhang, Fuhua Li

**Affiliations:** 1CAS and Shandong Province Key Laboratory of Experimental Marine Biology, Center for Ocean Mega-Science, Institute of Oceanology, Chinese Academy of Sciences, Qingdao 266071, China; mwsu@qdio.ac.cn (M.S.); yuanjb@qdio.ac.cn (J.Y.); zhangxiaoxi@qdio.ac.cn (X.Z.); fhli@qdio.ac.cn (F.L.); 2Laboratory for Marine Biology and Biotechnology, Qingdao National Laboratory for Marine Science and Technology, Qingdao 266237, China; 3College of Earth Science, University of Chinese Academy of Sciences, Beijing 100049, China

**Keywords:** *Litopenaeus vannamei*, insulin-like peptide, hemolymph glucose, homeostasis, glucose metabolism genes

## Abstract

Insulin-like peptide (ILP) has been identified in various crustaceans, but whether it has a similar function in regulating hemolymph glucose as vertebrate insulin is unclear. We analyzed the components of hemolymph sugar in the Pacific white shrimp, *Litopenaeus vannamei*, and investigated the changes of hemolymph glucose concentration and the expressions of ILP and glucose metabolism genes under different treatments. We found glucose was a major component of hemolymph sugar in shrimp. Starvation caused hemolymph glucose to rise first and then decline, and the raised hemolymph glucose after exogenous glucose injection returned to basal levels within a short time, indicating that shrimp have a regulatory mechanism to maintain hemolymph glucose homeostasis. In addition, injections of bovine insulin and recombinant *LvILP* protein both resulted in a fast decline in hemolymph glucose. Notably, RNA interference of *LvILP* did not significantly affect hemolymph glucose levels, but it inhibited exogenous glucose clearance. Based on the detection of glucose metabolism genes, we found *LvILP* might maintain hemolymph glucose stability by regulating the expression of these genes. These results suggest that ILP has a conserved function in shrimp similar to insulin in vertebrates and plays an important role in maintaining hemolymph glucose homeostasis.

## 1. Introduction

Carbohydrates are the main source of energy supply for animals, and a continuous supply of blood sugar can ensure normal function and survival. Glucose is a major component of blood sugar in vertebrates, and plasma glucose concentration can be maintained within a narrow range (approximately 5 mmol/L in humans), which is known as a physiological set point [1]. Insulin is a key hormone sustaining blood sugar stability. Studies have shown that insulin acts as a major metabolic hormone, activates PI3K/Akt signaling, and stimulates glucose intake by enhancing GLUT4 transport to the cell surface, thereby maintaining normal blood sugar levels for regular biological functions [2]. Disruption of insulin secretion causes metabolic disorders, such as hyperglycemia or hypertriglyceridemia, suggesting the important role of insulin in regulating blood sugar homeostasis [3,4].

The blood sugar of insects is different from that of vertebrates. In 1957, Wyatt and Kale isolated trehalose from the hemolymph of 10 species of insects and found trehalose accounted for more than 80% of the hemolymph sugar of insects [5]. After that, more evidence has shown that trehalose is the main component of insect hemolymph sugar, which is synthesized by fat body [6,7,8,9]. At the same time, it was also found that the hemolymph sugar of insects can be maintained at a relatively stable level. For example, the concentration of trehalose in hemolymph of the fifth-instar larvae of *Spodoptera exigua* is generally 2.31 ± 0.62 mmol/L [10].

Insulin-like peptide (ILP) is a member of the insulin superfamily, which was first found in insects and consists of N-terminal signal peptide (SP), B chain, non-conserved C peptide, and A chain. ILP can promote glucose metabolism, determine cell size and growth, accelerate metamorphosis, participate in immune responses, and affect reproduction and lifespan in insects [11]. In *Drosophila melanogaster*, ILP is expressed in the brain and regulates trehalose and glucose levels in the hemolymph; ablation of ILP-producing neurons in adult pars intercerebralis has been found to increase trehalose content [12,13]. Starvation of larval *S. exigua* and RNA interference (RNAi) of the ILP gene in the larval stages of *Apis mellifera*, *D. melanogaster*, and *Maruca vitrata* has been shown to increase the level of trehalose in hemolymph, while the injection of bovine insulin in *S. exigua* and *Antheraea pernyi* or overexpression of the ILP gene in *Drosophila* have decreased trehalose levels [10,12,13,14,15,16]. These results suggest that ILP plays a role in suppressing trehalose levels in insect hemolymph consistent with the function of insulin in vertebrates in regulating blood glucose levels.

Despite belonging to the same phylum (Arthropoda) as insects, there are few studies on the composition and regulation of hemolymph sugar in crustaceans. In 1968, Telford detected five carbohydrates, including glucose, maltose, trehalose, fructose, and galactose, using paper chromatography in the hemolymph of the American lobster *Homarus americanus*, of which glucose had the highest content [17]. After that, glucose, maltose, and maltotriose were also detected in the hemolymph of the purple shore crab *Hemigrapsus nudus*, but it lacked trehalose [18]. The main carbohydrates found in the hemolymph of sand crabs (*Emerita asiatica*) and sea cockroaches (*Ligia exotica*) were glucose and 6-phosphate glucose [19].

In 2003, Gallardo et al. isolated an insulin-like biological bioactive protein from the hepatopancreas of the Caribbean spiny lobster *Panulirus argus*; the protein interacted with insulin receptors and inhibited glucose metabolism in rat adipocytes [20]. Additionally, the first ILP in a decapod was identified in the Eastern Australian rock lobster *Sagmariasus verreauxi*, which was considered to be an orthologue of DILP7 in *D. melanogaster* [21]. Currently, ILP genes have been identified in various crustaceans, and these ILPs may be involved in the regulation of carbohydrate metabolism [22], immune responses [23], and gonadal development [11]. Early studies have found that an insulin injection had no effect on hemolymph glucose in the crayfish *Astacus trowbridgii* and the blue crab *Callinectes sapidus* [24], and a glucose injection in *H. americanus* did not increase immunoreactive insulin in hemolymph [25]. However, another study showed that an injection of bovine insulin in the blue crab *C. sapidus* reduced hemolymph glucose levels [26]. In the Pacific white shrimp *Litopenaeus vannamei*, increased glycogen content in the hepatopancreas was observed after 3 hours following an injection of bovine insulin and recombinant human insulin-like growth factor-I (rhIGF-I) [27]. These studies have shown that endogenous ILPs exist in crustaceans, and they may regulate glucose metabolism like insulin in vertebrates.

*L. vannamei* is the most economically important shrimp in the world, and glucose homeostasis is critical for its survival. Understanding the carbohydrate metabolism system of shrimp can help optimize farming techniques, but the related basic research is lacking. The composition of hemolymph sugar has not been determined, and the mechanisms of hemolymph sugar regulation are unclear. In this study, we cloned an ILP gene of *L. vannamei* and analyzed the relationship between ILP and hemolymph glucose changes under starvation treatment; injections of exogenous glucose, bovine insulin, and recombinant LvILP protein; and RNA interference of the LvILP gene in the shrimp. These studies will help in understanding the mechanisms of glucose metabolism and improve the healthy farming of shrimp.

## 2. Results

### 2.1. Glucose Is the Main Component of Shrimp Hemolymph Sugar

Six kinds of carbohydrates were detected in the hemolymph of *L. vannamei*, including four monosaccharides and two disaccharides, namely xylitol, inositol, glucose, D-Sorbitol, trehalose, and maltose. Among them, glucose concentration was the highest, reaching 1.73 mmol/L, followed by maltose, xylitol, trehalose, inositol, and D-sorbitol (Table 1). The concentration of glucose was 11.46 times that of maltose and 23.49 times that of trehalose, indicating that glucose is the main carbohydrate in the hemolymph of shrimp. The results are quite distinct from those of insects, in which trehalose is the major hemolymph sugar.

### 2.2. Starvation Stress Affected Hemolymph Glucose and the Expression of LvILP and Glucose Metabolism Genes

During a normal feeding period, the hemolymph glucose of *L. vannamei* was stable and sustained at around 0.99–1.64 mmol/L (Figure 1A). During four days of starvation, hemolymph glucose of the shrimp first rose, and it reached highest concentration at 12 h, which was 3.45 times higher than at 0 h. After that, the hemolymph glucose gradually decreased and returned to the initial level during 24–72 h.

During starvation, the expression of *LvILP* significantly decreased within 12 h after starvation and increased at 24 h (Figure 1B). A similar expression pattern was also found in the phosphofructokinase gene (*PFK*), whose expression decreased at 6 h and increased at 12 h, continuing until 72 h. The phosphoenolpyruvate carbon carboxylase gene (*PEPCK*) significantly increased at 12 h, decreased at 24 h, and returned to the initial level at 72 h. Most notably, the expression of the glycogen synthase gene (*GS*) decreased at 6 h, while the expression of the glycogen phosphorylase gene (*GP*) increased; their expressions both returned to the initial levels at 12 h, increased at 24 h, and decreased at 72 h. These results indicated that *LvILP* and the four glucose metabolism genes all respond to starvation stress.

### 2.3. Exogenous Glucose Was Rapidly Cleaned Up by Activating LvILP and Glucose Metabolism Genes

After exogenous glucose injection, the glucose content of the experimental group increased sharply within 5 min (Figure 2A) and was 5.28 times higher than that of the control group. After that, the glucose concentration of the experimental group began to decrease and returned to an equal level with the control group at 60 min. This indicated that the shrimp cleared the high concentration of exogenous glucose from the hemolymph in a short time.

The detection of the gene expressions showed that *LvILP* expression increased 10 min after glucose was injected, returned to the initial level at 20 min, and then gradually decreased (Figure 2B). At the same time, the expression levels of the glucose metabolism genes all changed, and the changes were relatively consistent. The expression levels of *GS*, *GP*, *PFK*, and *PEPCK* all increased at 60 min. It is noteworthy that the expression level of *GS* increased at 10 min, which was consistent with *LvILP*, while the expression level of *GP* did not change. The above results suggest that glucose metabolism of the shrimp was activated when clearing excess glucose in the hemolymph.

### 2.4. Bovine Insulin Reduced Hemolymph Glucose and Affected Glucose Metabolism Genes in L. vannamei

After fasting for 12 h and subsequent refeeding for 1 h, the hemolymph glucose concentration of the shrimp was at a high level (3.8 mmol/L) (Figure 3A). Within 10 to 30 min after injection with bovine insulin, the glucose concentration of the experimental group was lower than that of the control group, although not statistically significant. After 1 h, the hemolymph glucose concentration of the two groups both recovered to normal range (~1.5 mmol/L). This indicated that bovine insulin can rapidly reduce the high hemolymph glucose in shrimp after feeding.

The relative expression levels of *LvILP* and the key enzymes of glucose metabolism genes are shown in Figure 3B. At 10 and 30 min after bovine insulin injection, the expression of *LvILP* significantly decreased (*p* < 0.05). While the expression levels of *GS*, *PFK*, and *PEPCK* vastly increased at 30 min (*p* < 0.05), among them, *PEPCK* had the most increase. However, *GP* expression increased at 10 min and decreased at 30 min, indicating that exogenous insulin may inhibit glycogen degradation and promote glycogen synthesis and gluconeogenesis after feeding.

### 2.5. Recombinant LvILP Protein Accelerated the Decline of Hemolymph Glucose in L. vannamei

Recombinant LvILP protein was expressed and purified (Figure 4A,B). After the recombinant protein was injected for 30 min, the hemolymph glucose concentration of the experimental group was significantly lower than that of the control group (Figure 4C). The hemolymph glucose of all groups recovered to normal range within an hour. The results were similar to those of bovine insulin injection in this study, indicating that the recombinant LvILP protein accelerated the decline of hemolymph glucose. At 12 h after injection, the expressions of *GP*, *PFK*, and *PEPCK* all increased (Figure 4D). Among them, *PEPCK* had the most significant increase. These results were essentially consistent with those of bovine insulin injection, indicating that the effect of *LvILP* protein on hemolymph glucose and the glucose metabolism genes is similar to that of exogenous insulin.

### 2.6. RNA Interference of LvILP Inhibited the Clearance of Exogenous Glucose and Downregulated the Glucose Metabolism Genes

During RNAi of *LvILP* for 24 h, compared with the control group, there was no significant change in hemolymph glucose (Figure 5A). Gene expression detection showed that the expression of *LvILP* significantly reduced at 24 h, and the expression of *GS* and *PEPCK* decreased slightly, *GP* and *PFK* decreased significantly (*p* < 0.01) (Figure 5C), indicating that the knockdown of *LvILP* expression may inhibit glycogen metabolism and the glycolysis pathway. At this time point (24 h of RNAi of *LvILP*), glucose was injected to observe the ability of shrimp to remove exogenous glucose after *LvILP* had been knockdown. It was found that the hemolymph glucose concentration in the RNAi group was much higher than in the control group at 60 and 120 min after injection (Figure 5B). This indicated that the interference of the ILP gene can inhibit the clearance of exogenous glucose and affect hemolymph glucose regulation in shrimp.

## 3. Discussion

### 3.1. The Main Component of Hemolymph Sugar in Shrimp

Hemolymph sugar is an important source of energy and provides nutrients for every organ and life activity. The content of glucose in the hemolymph of *L. vannamei* was the highest, followed by maltose and trehalose, and the glucose content was 11.46 and 23.49 times that of maltose and trehalose, respectively, which indicated that glucose is the most important carbohydrate in shrimp hemolymph. This is consistent with the results that have been previously found in the lobster *H. americanus* and the crabs *Cancer magister* and *H**. nudus* [17,18]. Correspondingly, in insects, trehalose synthesized in the fat body is the dominant carbohydrate in insect hemolymph, while glucose is at a lower level [5,6,28,29].

Trehalose is a typical stress-responsive or anti-stress metabolite. It can stabilize cellular membranes and protect protein structure by replacing water molecules and facilitating cell vitrification to deal with abiotic stresses [6,7,8,9]. Insects are terrestrial arthropods and face variable external conditions, such as cold, heat, dehydration, ultraviolet radiation, and even extreme environments [30,31]. Therefore, in the evolutionary process, insects chose trehalose as the major component of hemolymph sugar to resist harsh environments. However, as aquatic animals, crustaceans have a relatively stable physical environment, so it may be that they did not choose trehalose and, instead, use glucose as the major component of hemolymph sugar.

Maltose is an intermediate sugar formed by the action of amylase-catalyzed hydrolysis of starch; it provides energy for the body. Xylitol, inositol, and D-sorbitol are three sugar alcohols. Their contents are very low in shrimp hemolymph, and they may mediate cell signal transduction, participate in osmoregulation, and affect insulin and lipid concentrations in the blood [32,33].

### 3.2. Shrimp Have a Mechanism to Maintain Hemolymph Glucose Homeostasis

Hemolymph glucose balance is vital for maintaining homeostasis. In this study, the hemolymph glucose was at about 0.99–1.64 mmol/L during a feeding period of 4 days in *L. vannamei*; during starvation, hemolymph glucose slowly increased within the first 12 h, and then decreased gradually to the initial level at 72 h (Figure 1A). The increase of hemolymph glucose in the early stages of starvation was probably a stress response. For example, hemolymph glucose of the American lobster *H. americanus* maintained a high level during short-term starvation and decreased after long-term fasting [17]. In this study, the shrimp showed a phase of elevated hemolymph glucose similar to that of the American lobster under starvation treatment. A subsequent exogenous glucose injection experiment also showed that shrimp could quickly turn over glucose from a very high level to a normal range within 1 h (Figure 2A). An injection of H_2_O or phosphate buffered saline (PBS) in the control group caused a small increase in hemolymph glucose of the shrimp, and the glucose level returned to normal within a short time (Figure 2A, Figure 4A and Figure 5A). We speculate that, regardless, starvation, exogenous glucose, exogenous insulin (recombinant ILP), double-stranded *ILP*, and injection of a solvent are all stimulants to shrimp and affect hemolymph glucose balance. However, the body will eventually regulate hemolymph glucose at a normal level. These results imply that shrimp have a mechanism to maintain hemolymph glucose homeostasis.

### 3.3. LvILP May Flexibly Regulate Hemolymph Glucose in Multiple Ways

In vertebrates, glucose homeostasis is the result of the combined effect of neuromodulation and humoral regulation, in which insulin and glucagon play major roles in regulating blood sugar balance. Crustaceans do not have the above two hormones, but instead have their homologs, ILP and crustacean hyperglycemic hormone (CHH). However, the regulation mechanisms of hemolymph glucose homeostasis by ILP and CHH remain poorly understood in crustaceans.

In this study, we found that the expression of *LvILP* was inhibited when the hemolymph glucose concentration increased at the initial stage of starvation (6–12 h), and it then gradually increased when hemolymph glucose concentration decreased (24–72 h) (Figure 1B). A similar situation exists in *S. exigua*, in which starvation resulted in a significant (more than two-fold) increase in trehalose levels within 48 h, but almost no *SeILP1* was transcribed at that time [7]. This indicates that the *LvILP* may not respond noticeably to the increase of hemolymph sugar caused by stress at the initial stage of starvation, and its response may even be suppressed so that hemolymph sugar change is not obvious. On the other hand, it was reported that knocked-down ILP genes enhanced starvation resistance capability and prolonged survival time in the brown planthopper *Nilaparvata lugens* and *D. melanogaster*, respectively [34,35]. In our study, the expression of *LvILP* increased in a short time after injecting high concentrations of glucose (Figure 2B). These results indicated that LvILP is involved in the regulation of hemolymph glucose in three aspects: During the short-term starvation, the expression of *LvILP* decreased to elevate the hemolymph glucose level. When the hemolymph glucose decreased due to long-term starvation, the expression of *LvILP* increased, which may have been to strengthen the regulation of glucose metabolism and maintain the lowest hemolymph glucose level. When hemolymph glucose concentrations rose rapidly, the *LvILP* was highly expressed and quickly removed excess glucose. Therefore, we deduced that the ILP gene flexibly responds to hemolymph glucose changes according to different conditions.

Additionally, the injection of bovine insulin accelerated the conversion rate of hemolymph glucose in the experimental group, which indicated that bovine insulin can regulate hemolymph glucose in shrimp. This is similar to the results of exogenous insulin injections in the tussah *A. pernyi*, the mouth crab *Neohelice granulata*, and the barnacles *Balanus nubilus* [16,36,37]. In our experiment 10–30 min after bovine insulin injection, the expression of *LvILP* significantly reduced (Figure 3B), indicating that exogenous insulin had a certain inhibitory effect on *LvILP*. Further research found that, after recombinant LvILP protein injection, the changes in hemolymph glucose were essentially the same as bovine insulin. This suggests that the recombinant LvILP protein also declined hemolymph glucose.

Notably, after 24 h of a dsLvILP injection, there was no significant difference in hemolymph glucose concentration between the experimental and control groups. The same result was observed in *Macrobrachium rosenbergii* [22], but it was different from the hemolymph glucose increase in insects after their ILP gene was interfered [10,14,15,16]. We speculate that the reason may be that there is a strong hemolymph glucose maintenance mechanism in crustaceans. There may be other factors regulating hemolymph glucose; even if the expression of the ILP gene is significantly decreased, hemolymph glucose can still be preserved steadily. Another finding was that when glucose was injected to the *LvILP* RNAi shrimp (the relative expression level of *LvILP* was 10% of the control), the hemolymph glucose concentration of the experimental group was significantly higher than the control group at 60 and 120 min, indicating that *LvILP* RNAi can decelerate the removal of excess hemolymph glucose in the shrimp. The above studies suggest that LvILP has the same ability to negatively regulate hemolymph glucose as insulin in vertebrates.

### 3.4. LvILP Regulates Glucose Metabolism

Hemolymph glucose homeostasis is closely relevant to the process of glucose metabolism. In the beetle *Harmonia axyridis* after starvation for 12–24 h, the expression of *HaGP* was significantly higher than at 8 h [38]. In *S. exigua,* the mRNA levels of *SpoexGS* and *SpoexGP* in the experimental group starved for 24 h were also higher than at 6 h and 12 h [39]. In this study after 6 h of starvation, glycogenolysis was activated, and glycogen was broken down into glucoses so that shrimp hemolymph glucose increased gradually, and glucose metabolism and glycogen synthesis were inhibited. At 12 h of starvation, both glycogen and glucose metabolism were activated, especially *PEPCK*, which increased significantly, indicating that shrimp may obtain new glucose reinforcement through gluconeogenesis. Coupled with glycogenolysis, hemolymph glucose reached a peak at this time. At 24 h of starvation, the expression levels of *GS*, *GP*, and *FPK* all rose to peak, indicating that glycogen metabolism and the glycolytic pathway were highly active. At 72 h of starvation, glycogen metabolism was weakened, indicating that long-term starvation led to a large amount of glycogen being exhausted. The above results suggest that the shrimp first provided glucose through glycogen metabolism and then replenished glucose through gluconeogenesis and gained energy through glycolysis during starvation. Although gluconeogenesis implies large energy consumption, maintaining glucose homeostasis is more important for survival.

At 10 min after glucose injection in the shrimp, hemolymph glucose concentration increased. Simultaneously, *LvILP* and *GS* had high expression levels, and the synthesis of glycogen was rapidly activated. This indicated that shrimp can first store exogenous glucose as glycogen. At 60 min after glucose injection, hemolymph glucose returned to normal range and *LvILP* expression decreased; however, the expressions of the four key genes of glucose metabolism were high. These results indicate that the ILP gene responded significantly and rapidly to high hemolymph glucose concentrations. These glucose metabolism genes are located downstream of hemolymph glucose regulation. This is similar to glucose injection in *M. rosenbergii*, *Oreochromis niloticus*, *Carassius gibelio*, and *Ctenopharyngodon idella* [22,40,41,42].

At 30 min after bovine insulin injection, the expression levels of *GS*, *PFK*, and *PEPCK* increased significantly, indicating that bovine insulin promoted the process of glucose metabolism. At 12 h after recombinant LvILP protein injection, the expressions of *GP*, *PFK*, and *PEPCK* all increased, which is essentially consistent with the changes after bovine insulin injection in this study but contrary to the results after *LvILP* RNAi. At 24 h of *LvILP* RNAi, the expressions of *GS* and *PEPCK* decreased slightly, while *GP* and *PFK* decreased significantly. A similar situation was also found in *M. rosenbergii* [22]. All these results indicate that LvILP can regulate the expression of glucose metabolism genes to maintain blood glucose homeostasis. In mammals, insulin and IGF-1 can inhibit the activity of GSK3β through the PI3K signaling pathway, resulting in significant dephosphorylation and activation of GS, which both play a key role in the regulation of glycogen synthesis [43,44]. We speculated that LvILP may regulate the glucose metabolism genes through a conserved insulin pathway, but the mechanisms need to be further studied.

## 4. Materials and Methods

### 4.1. Experimental Animals

Healthy shrimp (*L. vannamei*) with a body weight of 4–5 g were cultured in the Aquarium Building of the Institute of Oceanology, Chinese Academy of Sciences (Qingdao, China) for at least one week before experiments. 20 shrimp were kept in each rearing tank (700 mm × 500 mm × 450 mm) with sufficient oxygen, a temperature of 25 ± 0.2 °C, a salinity of 0.3%, and a pH of 7.5 ± 0.1. The seawater was changed once a day. Food pellets (Dell Feed Company, Yantai, China) were provided 3 times a day at 9 a.m., 3 p.m., and 9 p.m. No rare or endangered animals were used in this study.

### 4.2. Identification and Cloning of LvILP Gene

Sequences annotated as ILP genes were collected from the *L. vannamei* genome and transcriptome data in our laboratory. Several sequences with high similarity to ILP genes from related species were selected by blastx alignment on the NCBI website (https://blast.ncbi.nlm.nih.gov/Blast.cgi, accessed on 2 January 2021), and their amino acid sequences were predicted on the ExPASy website (https://web.expasy.org/translate/, accessed on 2 January 2021). Then, the conserved domains of these amino acid sequences were predicted using SMART (http://smart.embl-heidelberg.de/, accessed on 2 January 2021). Finally, an LvILP gene with a complete sequence was identified.

Total RNA was extracted from three shrimp eyestalk using RNAiso Plus Reagent (Takara Bio Inc., Kyoto, Japan) according to manufacturer protocol. The concentration and quality of the RNA were detected by a Nanodrop^TM^ 2000 Spectrophotometer (Thermo Fisher Scientific, MA, USA) and 1.5% (*w*/*v*) agarose gel electrophoresis. Then, the RNA was reverse-transcribed to cDNA using a PrimeScript™ RT Reagent Kit (Takara Bio Inc., Kyoto, Japan). To amplify the LvILP gene from the cDNA, primers were designed using Primer3 (https://www.primer3plus.com/cgi-bin/dev/primer3plus.cgi, accessed on 15 December 2021) (Appendix A), and the target products were detected by 1.5% (*w*/*v*) agarose gel electrophoresis. After verification of band size, PCR products were sequenced at Sangon Biotech Co., Ltd. (Shanghai, China). The obtained sequences were confirmed by alignment with LvILP genes from the genome and transcriptome, and, finally, the complete sequence of the LvILP gene was cloned into a pMD19-T vector (Takara Bio Inc., Kyoto, Japan) for use.

### 4.3. Detection of Hemolymph Sugar Components in L. vannamei

Twelve healthy shrimp were randomly selected. About 1 mL hemolymph was collected from four shrimp and mixed with an equal volume of sodium citrate anticoagulant (G-CLONE, Beijing) as a sample, and three replicate samples were collected. Finally, all hemolymph samples were frozen in liquid nitrogen and stored at −80 °C in a freezer.

A total of 500 μL mixture consisting of methanol, isopropanol, and water (3:3:2 by volume) was added to 50 μL of the hemolymph sample. After spinning for 3 min and sonication for 30 min, the extract was centrifuged at 14,000 rpm for 3 min at 4 °C. The supernatant (50 μL) was mixed with 20 μL of the internal standard (ribitol, 100 μg/mL) and evaporated under nitrogen flow. The evaporated samples were freeze-dried in a lyophilizer and then further derivatized. The derivatized mixture was diluted to the appropriate concentration and analyzed using GC-MS.

An Agilent 7890B Gas Chromatograph (GC) System (Agilent Technologies, Palo Alto, CA, USA) coupled with an Agilent 7000D Triple Quadrupole GC/MS (Agilent Technologies Palo Alto, CA, USA) with a DB-5ms column (30 m × 0.25 mm × 0.25 μm, J&W Scientific, Inc., Folsom, CA, USA) was used to detect carbohydrates in the shrimp hemolymph. Helium was the carrier gas at a flow rate of 1 mL/min. Sample loading was performed in a split ratio of 3:1 with an injection volume of 3 μL. After detection, the sugar content was calculated according to the standard.

### 4.4. Starvation Experiment

To detect changes in hemolymph glucose and the expressions of *LvILP* and glucose metabolism genes in shrimp during starvation, sixty shrimp were randomly divided into two groups after fasting for 12 h to eliminate initial hemolymph glucose difference caused by individual diet. The control group shrimp were fed normally, and the experimental group underwent starvation. Samples were collected at 0, 6, 12, 24, 48, 72, and 96 h after treatment.

At each time point, hemolymph samples from 3 shrimp were collected and mixed with an equal volume of anticoagulant into one sample. Each sample was placed on ice for subsequent detection of glucose content. At the same time, the eyestalk, hepatopancreas, and intestines were collected. The same tissues from three shrimp were mixed into one sample, and three biological replicates were collected. The tissue samples were quickly put into liquid nitrogen for subsequent detection of gene expression. Finally, all samples were stored at −80 °C in a freezer.

The glucose content of each sample was determined using a glucose micro-detection kit (Solarbio Science & Technology Co. Ltd., Beijing, China), and the experiment was carried out according to manufacturer instructions. Total RNA was extracted from the eyestalks and reverse-transcribed into cDNA. Primers for *LvILP* and the genes of the glucose metabolism key enzyme, including *GS*, *GP*, *PFK*, and *PEPCK*, were designed on the Primer3 website. The relative expression levels of target genes in each sample were detected with an Eppendorf Mastercycler ep realplex system (Eppendorf, Hamburg, Germany) using a Quant One-Step Probe RT-qPCR Kit (SYBR Green) with 18 S rRNA as an internal reference. The RT-qPCR reaction program was as follows: pre-denaturation at 95 °C for 2 min; 40 cycles of denaturation at 95 °C for 20 s; annealing at 55 °C or 56 °C for 20 s; extension at 72 °C for 20 s; and extension at 72 °C for 10 min (Appendix A). Each sample had four technical replicates. The melting curves were used as standards for PCR product specificity. The relative expression level of each gene was calculated with the 2^−ΔΔCt^ method [45].

### 4.5. Glucose Injection

To detect the changes in hemolymph glucose and the expressions of *LvILP* and glucose metabolism genes in shrimp after injection with exogenous glucose, 120 shrimp were fasted for 12 h and divided into two groups. The experimental group was injected with glucose at 0.75 g/kg body weight, and the control group was injected with an equal volume of H_2_O. A 50 μL volume microsyringe was used to inject the solvent into the side muscle of the third abdomen of the shrimp. Samples were collected at 0, 5, 10, 20, 60, 180, and 420 min after injection. The experimental steps for sampling and detection were the same as those in Section 4.4.

### 4.6. Bovine Insulin Injection

To determine whether exogenous insulin had a regulatory role on hemolymph glucose and affected the expressions of glucose metabolism genes in shrimp, 90 shrimp were fasted for 12 h and then refed with commercial food. After 1 h, they were divided into two groups. The experimental group shrimp were injected with bovine insulin (Solarbio Science & Technology Co. Ltd., Beijing, China) at 1.5 IU/kg body weight, and the control group was injected with an equal volume of H_2_O. Samples were collected at 0, 10, 30, 60, and 120 min after injection. The experimental steps for sampling and detection were the same as those in Section 4.4.

### 4.7. Injection of Recombinant LvILP Protein

A pMAL-c5X vector (TIANZA, Beijing, China) was used to construct an LvILP recombinant expression system. The recombinant *LvILP*-pMAL-c5X-His plasmid was transfected into *E. coli* TransB (DE3) chemically competent cells (TransGen Biotech Co. Ltd., Beijing, China) and continuously cultured in Luria-Bertani (LB) medium containing 100 μg/mL ampicillin at 37 °C and 220 rpm. Isopropyl-beta-D-thiogalactopyranoside (IPTG) with a final concentration of 1 mM was added to the medium at an OD_600_ of 0.4–0.6, and the cells were incubated at 18 °C for 24 h to induce sufficient fusion protein. Subsequently, the cells were collected by centrifugation at 8000 rpm for 5 min, resuspended in PBS solution, and sonicated at 4 °C for 30 min to release the intracellular fusion protein. Fusion protein in the cell lysate was detected by 12.5% sodium dodecyl sulfate-polyacrylamide gel (SDS-PAGE) electrophoresis, and the target protein was purified using a durable immobilized metal affinity chromatography (IMAC) resin (TALON^®^ Metal Affinity Resin, Takara Bio Inc., Kyoto, Japan). Finally, the obtained recombinant LvILP protein was identified by SDS-PAGE and the concentration was detected using a BCA Protein Quantification Kit (Vazyme Biotech Co., Nanjing, China); it was then stored at 4 °C. Maltose binding protein (MBP) was concurrently induced and purified as a negative control.

To study whether the recombinant LvILP protein had similar functions to insulin, 120 healthy shrimp were randomly selected and divided into 3 groups: the PBS control group, the MBP injection group, and the recombinant LvILP injection group. The treatment method was the same as that in Section 4.6. The shrimp were fasted for 12 h and then refed with commercial food. After 1 h, they were injected with 10 µL PBS, 1 nmol MBP protein (dissolved in 10 µL PBS), and 1 nmol recombinant LvILP protein (dissolved in 10 µL PBS), for the PBS control, MBP injection, and recombinant LvILP groups, respectively. Samples were collected at 0, 30, 60, 120, and 360 min after injection. The sampling and experimental steps were the same as those in Section 4.4.

### 4.8. RNA Interference of LvILP Gene

The bacteria with recombinant pMD19-T vector plasmid and the LvILP gene were recultured, and plasmid DNA was extracted using an E.Z.N.A.^®^ Plasmid DNA Mini Kit I extraction kit according to manufacturer instructions (Omega Bioservices, Norcross, GA, USA). Using 100 ng of plasmid DNA as a template and primer containing T7 promoter, the *LvILP* was amplified. The PCR product was purified using a SteadyPure PCR DNA Purification Kit (Accurate Bio-Medical Technology Co., Ltd., Changsha, China) and then used as a template to synthesize the dsRNA of the *LvILP* with a double-stranded synthesis kit (TranscriptAid T7 High Yield Transcription Kit, Thermo Fisher Scientific, Beijing, China). Simultaneously, the dsRNA of the enhanced green fluorescent protein (EGFP) gene was synthesized as a negative control. The concentration and quality of the synthesized dsRNA were detected using a Nanodrop^TM^ 2000 Spectrophotometer (Thermo Fisher Scientific, MA, USA) and 1.5% (*w*/*v*) agarose gel electrophoresis.

Sixty healthy shrimp were equally divided into three groups (the PBS group, the dsEGFP group, and the dsILP group) and injected with 10 µL PBS, 10 µL dsEGFP (6 μg/10 µL), and 10 µL dsILP (6 μg/10 µL), respectively. Samples were collected after injection at 0, 2, 6, 12, and 24 h for subsequent detection of hemolymph glucose and gene expression. Furthermore, we examined the ability of the shrimp to clear exogenous glucose after the ILP gene was knocked down. After RNAi of *LvILP* for 24 h, sixty shrimp were injected with glucose at 0.75 g/kg body weight. The samples were collected at 0, 10, 30, 60, and 120 min after injection. The sampling and experimental steps were the same as those in Section 4.4.

### 4.9. Data Analysis

Data were analyzed by one-way ANOVA using SPSS 22.0 software (https://www.ibm.com/products/spss-statistics, accessed on 19 January 2022), and Bonferroni and Tamhane’s T2 post-hoc tests were used for group comparison. The graphs were plotted using GraphPad Prism7 (https://www.graphpad.com/, accessed on 19 January 2022).

## 5. Conclusions

In this study, we analyzed the relationship between hemolymph glucose changes and ILP in *L. vannamei*. Our results showed that glucose is the main component in hemolymph sugar in shrimp. Shrimp have a regulatory mechanism to maintain hemolymph glucose homeostasis. Injections of recombinant LvILP protein and bovine insulin accelerated the decline rate of glucose in hemolymph, indicating that LvILP may have hemolymph hypoglycemic effects similar to vertebrate insulin. The glucose change was closely related to the expression of glucose metabolism genes. A recombinant LvILP protein injection and RNA interference of *LvILP* affected the expressions of these genes, which suggests that ILP may regulate hemolymph glucose levels through the downstream glucose metabolism genes. However, we also found that the level of glucose in hemolymph did not change significantly after *LvILP* was interfered, indicating that there may be other factors regulating hemolymph glucose, and further research about the mechanisms of hemolymph glucose homeostasis in shrimp is needed. These results provide important clues for studying the function of the crustacean ILP gene and its mechanism in regulating glucose metabolism.

## Figures and Tables

**Figure 1 ijms-23-03268-f001:**
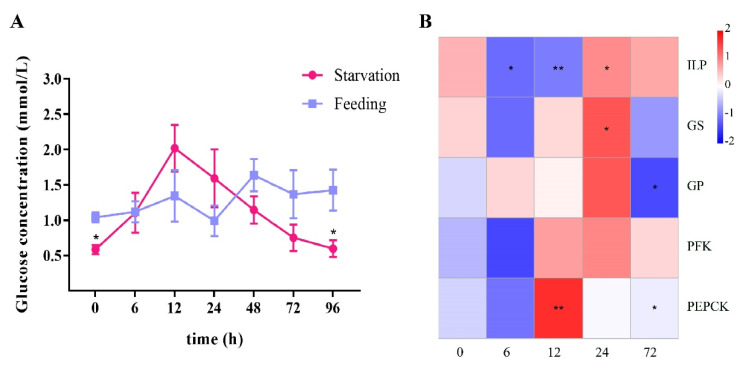
Effects of starvation treatment on hemolymph glucose and the expressions of *LvILP* and glucose metabolism genes in *L. vannamei*. (**A**) The variation curve of hemolymph glucose concentration during 4 days of starvation; error bars indicate the mean ± SD of three replicate samples. (**B**) Heatmap of the relative expression levels of *LvILP*, *GS*, *GP*, *PFK*, and *PEPCK* at different time points relative to the control group. The redder color indicates a higher expression level, and the bluer indicates a lower expression level. The numbers at the bottom represent the starvation time (h). The expression levels of the genes were detected by RT-qPCR, and 18 S rRNA was used as an internal reference. Statistical significances are shown as * *p* < 0.05 and ** *p* < 0.01.

**Figure 2 ijms-23-03268-f002:**
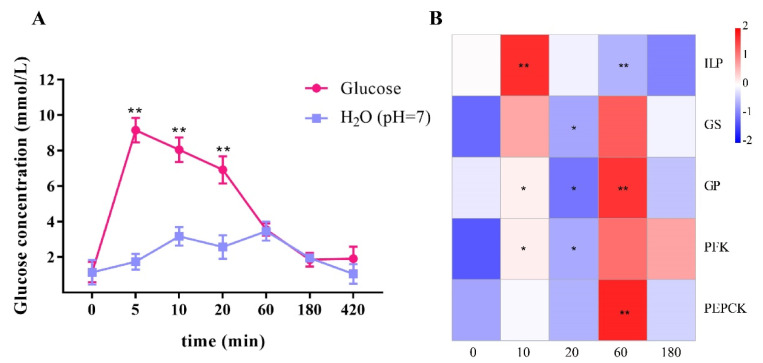
Effects of glucose injection on hemolymph glucose and the expressions of *LvILP* and glucose metabolism genes. (**A**) The variation curve of hemolymph glucose concentration of *L. vannamei* within 420 min of glucose injection; error bars indicate the mean ± SD of three replicate samples. (**B**) Heatmap of the relative expression levels of *LvILP*, *GS*, *GP*, *PFK*, and *PEPCK* genes at different time points relative to the control group. The redder color indicates a higher expression level, and the bluer indicates a lower expression level. The numbers at the bottom represent the treatment time (min). The expression levels of the genes were detected by RT-qPCR, and 18 S rRNA was used as an internal reference. Statistical significances are shown as * *p* < 0.05 and ** *p* < 0.01.

**Figure 3 ijms-23-03268-f003:**
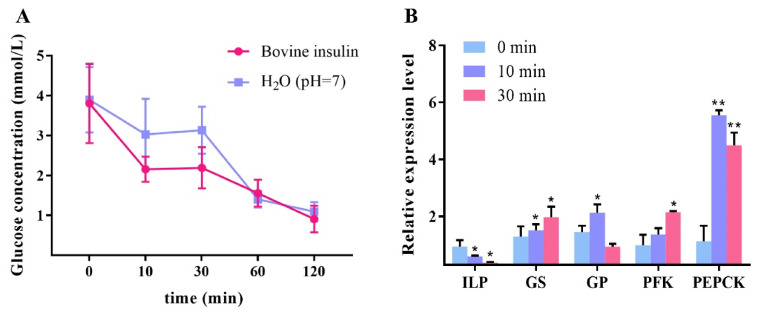
Effects of bovine insulin injection on hemolymph glucose and the expressions of *LvILP* and glucose metabolism genes. (**A**) The variation curve of hemolymph glucose concentration of *L. vannamei* after in vitro injection with bovine insulin. (**B**) Changes of the relative expression levels of *LvILP*, *GS*, *GP*, *PFK*, and *PEPCK* at 0 min, 10 min, and 30 min after injection with bovine insulin. The expression levels of the genes were detected by RT-qPCR, and 18 S rRNA was used as an internal reference. Error bars indicate the mean ± SD of three replicate samples. Statistical significances are shown as * *p* < 0.05 and ** *p* < 0.01.

**Figure 4 ijms-23-03268-f004:**
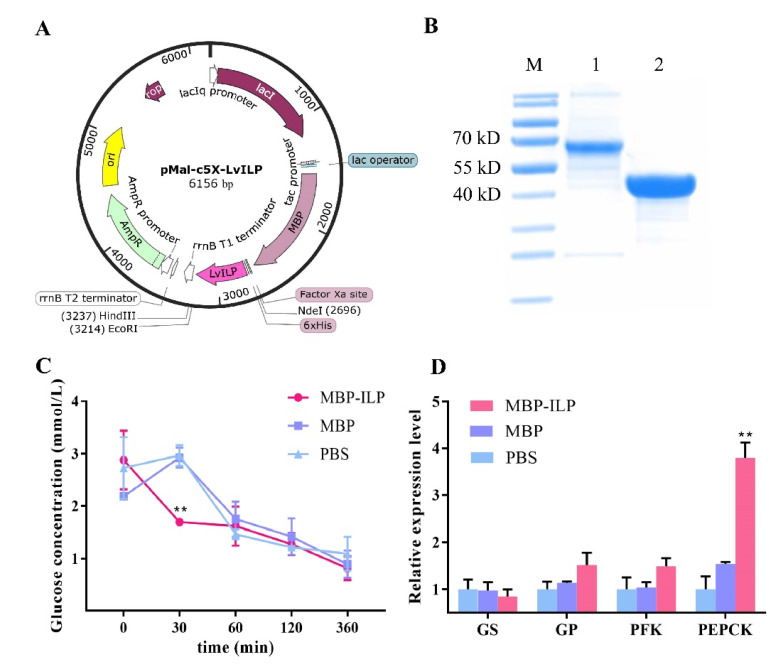
Effects of the recombinant maltose-binding protein (MBP)–LvILP protein injection on hemolymph glucose and the expressions of glucose metabolism-related genes. (**A**) The recombinant plasmid map of pMal-c5X-LvILP. (**B**) Sodium dodecyl sulfate-polyacrylamide gel electrophoresis (SDS-PAGE) electrophoresis of the recombinant MBP–ILP protein and the MBP protein. ‘M’ indicates a PageRuler^TM^ prestained protein ladder (Thermo Fisher Scientific, 26619); ‘1’ indicates purified MBP-ILP protein (63 kD); and ‘2’ indicates purified MBP protein (45 kD). (**C**) The variation curve of hemolymph glucose concentration within 6 h after recombinant LvILP protein injection. (**D**) The relative expression levels of *GS*, *GP*, *PFK*, and *PEPCK* at 12 h after injection of recombinant ILP protein. The gene expression levels were detected by RT-qPCR, and 18 S rRNA was used as an internal reference. Error bars indicate the mean ± SD of three replicate samples. Statistical significances are shown as ** *p* < 0.01.

**Figure 5 ijms-23-03268-f005:**
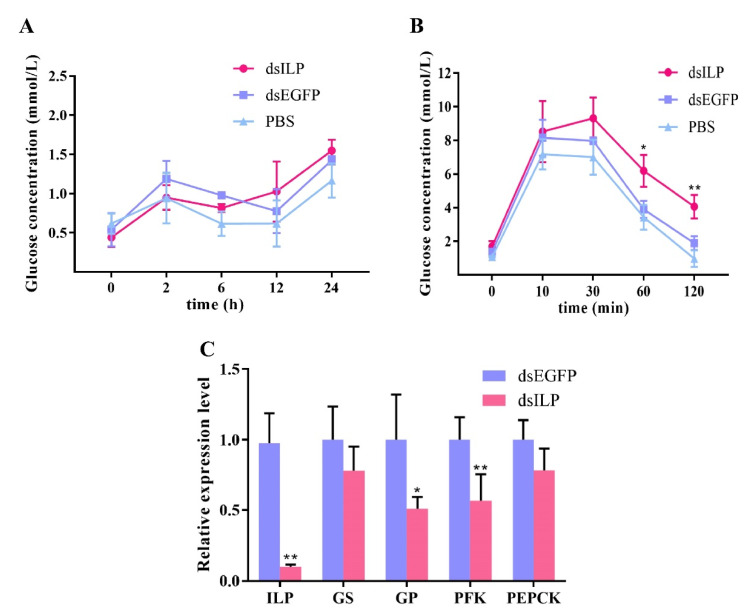
Effects of *LvILP* interference on hemolymph glucose and the expressions of glucose metabolism genes. (**A**) The variation curve of hemolymph glucose concentration within 24 h after RNAi of *LvILP*. (**B**) The variation curves of hemolymph glucose concentration within 2 h after glucose was injected into the *LvILP* RNAi shrimp. (**C**) The expression levels of *LvILP* and the glucose metabolism genes at 24 h after dsLvILP injection. Error bars indicate the mean ± SD of three replicate samples. Statistical significances are shown as * *p* < 0.05 and ** *p* < 0.01.

**Table 1 ijms-23-03268-t001:** Carbohydrate components in hemolymph of *L. vannamei*.

Carbohydrate	Substance Class	Concentration (mmol/L)
Xylitol	Monosaccharide	0.109
Inositol	Monosaccharide	0.053
Glucose	Monosaccharide	1.726
D-Sorbitol	Monosaccharide	0.049
L-Rhamnose	Monosaccharide	-
L-Fucose	Monosaccharide	-
D-Galactose	Monosaccharide	-
D-Fructose	Monosaccharide	-
D-Arabinose	Monosaccharide	-
Trehalose	Disaccharide	0.073
Maltose	Disaccharide	0.151
Sucrose	Disaccharide	-
Lactose	Disaccharide	-

“-” indicates that the corresponding carbohydrate was not detected.

## Data Availability

The data presented in this study are available on request from the corresponding author.

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
