# Peer review of "The Role of Insulin-like Peptide in Maintaining Hemolymph Glucose Homeostasis in the Pacific White Shrimp Litopenaeus vannamei"

_ijms, 2022, doi:10.3390/ijms23063268_

Round 1

Reviewer 1 Report

Su et al. conducted an interesting study on the role of ILP on the glucose homeostasis of Pacific White Shrimp. I found the study to be interesting. The manuscript needs revision for some points that I marked in the pdf version of this manuscript. 

Reviewer 2 Report

This manuscripts report experimental data suggesting a role for insulin like peptide from shrimp similar to that of insulin. The study finds that the expression of the protein is regulated by hemolymph glucose levels, and that injection of the protein or knock-down of the protein using RNAi affects glucose levels in the animal. Overall I find the manuscript well written and interesting, and the methods used are sound. However, some revisions will be needed, especially regarding more information about the experimental methods.

The term blood is used for the shrimp circulatory fluid,  but I believe that hemolymph is the correct term.   

It would be nice if all experimental data could be shown in a supplementary table.

The language is overall good, but there are many smaller grammatical errors and incorrect use of prepositions.

Fig 1: starvation is misspelled in legend.

Figs 1A + 2A + 3A/B + 4C/D + 5A/B/C: Are error bars showing standard deviation or standard error? This should be added to the figure caption.

Fig1B: Protein abbreviations should be spelled out first time they appear. As is, this is only done in the materials and methods section, which is the last section.

Fig. 2B: Why do glucose levels increase after injection of water? Is this due to stress caused by the injection?

Line 155 – 159: It is stated that the reduction in glucose concentration upon injection of bovine insulin is not significant, but still the conclusion is that insulin can reduce blood glucose levels. So what exactly do you mean by “not significant”? If you mean statistically significant, the conclusion is false, but maybe you just mean “not very pronounced”? But in that case the difference should be tested statistically!

Line 268 – 269: I do not quite follow this argument. If blood glucose is very low, how can LvILP maintain the stability of blood glucose? Could it be that glucose is released from storage organs (hepatopancreas?) and needs LvILP to be absorbed by the body cells? Please explain in more details.

Lin 290: Even though injection of dsLvILP caused an increase in glucose level, this increase was not very pronounced. Is this because the RNAi knockdown is not 100 % efficient, or could it be because other mechanisms exist to regulate glucose levels? What is known from other studies about this?

Line 351 + 391 + 398: It needs to be explained why the eyestalk was used. I assume LvILP is produced in the eyestalk glands, but this needs to be explained.  

Line 401: You need to give all sequences of primers used in the study.

Line 403: Concentrations of the components of the RT-qPCR reactions should be given.

It is also not evident if all organs collected were used for gene expression measurements, and if so, how were the final values obtained? Were tissues pooled before RT-qPCR, or were average Ct values calculated after PCR? 

Line 472: Were any post-hoc tests used after doing the ANOVA? If not, how can you determine statistically significance of single factors in the ANOVA?

For the injection experiments, please give more details about what equipment was used for injections, and where in the animals injections were given.

Round 2

Reviewer 1 Report

The authors have made substantial progress in the revised version of the manuscript. They have addressed all of my comments very well. I congratulate the authors for this excellent work.